# The impact of internet usage preferences on labor income: Evidence from China

**Kefeng Yuan** [1,2], **Xiaoxia Zhang** [1,2]*

1 Institute of Economics and Management, Ningde Normal University, Ningde, Fujian Province, China,
2 Institute of Education and Arts, Tomsk State University, Tomsk, Tomsk State, Russia

* t0917@ndnu.edu.cn

**Data Availability Statement:** All CGSS2017 files are available from the CGSS database (http://www.cnsda.org/).

**Funding:** This work was supported by grants from the National Social Science Foundation of China (No.22BJY045), Fujian Provincial Science and

## Abstract

### Background

The widespread application and iterative updating of computers and Internet communication technologies have not only increased productivity and enhanced intra- and inter-enterprise collaboration, but have also led to significant changes in the labor market and residents' labor income. In the digital era, accepting digital technology and possessing a certain degree of digital literacy have become the necessary abilities for people to survive and develop. However, the differences in digital literacy caused by individual differences will inevitably bring about a series of chain reactions. Therefore, it is necessary to study the subtle impact of Internet usage preference on residents' labor income in the context of digital transformation to promote digital equity.

### Objective

This study aims to empirically analyze micro-level survey data to reveal the impact of individual differences in internet usage preferences on their labor income. The findings provide theoretical references for government policy formulation and individual development.

### Methods

A function model was established to analyze the impact of individual internet usage preferences on labor income. Relevant data from the authoritative Chinese General Social Survey (CGSS2017) were selected, and empirical analyses for significance, heterogeneity, and robustness were conducted using the ZINB and CMP models in the Stata statistical software.

### Conclusion

(1) Higher Internet Usage Frequency (IUF) increases the likelihood of higher income. (2) Engaging in Online Social Networking (OSN) helps in accumulating social capital, leading to higher labor income. Meanwhile, participating in Online Entertainment (OE) relieves work and life stresses, thereby increasing labor income. Proficiency in Accessing Online Information (AOI) is associated with higher labor income, while frequent involvement in Online Business (OB) is correlated with higher personal income. Additionally, the Marginal utility of

Technology Department (No. 2023R0071) and
Ningde Normal University (No. 2022FZ01).

**Competing interests:** The authors have declared
that no competing interests exist.

these internet usage preferences indicate that OB > AOI > OSN > OE. (3) Individual variations in physical, psychological, and social characteristics significantly influence the labor income effects of internet usage preferences. (4) There are substantial differences in the labor income effects of internet usage preferences between urban and rural areas and across different regions. (5) Education attainment has a positive mediating effect on the labour income effect of individual Internet use preferences, and enhancing residents' digital literacy has a positive effect on increasing their labour income and alleviating inequality in digital gains. (6) The popularity of Internet technology is the background that triggers an individual's Internet use, and the acceptance of a particular Internet technology is catalyzed by an individual's perception of the value and difficulty of mastering that technology; an individual's biased learning or proficiency in a particular Internet technology in order to gain higher competitiveness and value in the labour market is an important internal driving force.

## 1. Introduction

As computer technology continues to advance, it enables people to process data and perform complex calculations more quickly and accurately, thereby increasing work efficiency. The Internet, on the other hand, establishes a global network that enables people to exchange information and share resources electronically (Berners-Lee T., 1999) [1]. Currently, the rapid development and popularisation of the new generation of Internet communication technologies represented by big data, cloud computing, Internet of Things, artificial intelligence, and 5G have not only greatly improved the efficiency of information and resource sharing, but also changed people's work and life style. However, differences in people's physical, psychological, and social characteristics have led to specific preferences or tendencies that people exhibit when using computers and the Internet (Roos J.M. & Kazemi A., 2021; Amichai-Hamburger Y. & Vinitzky G., 2010) [2, 3]. These Internet usage preferences may relate to an individual's preference for specific types of content, features, or services, and significantly impact an individual's work and life (Amiel T. & Sargent S., 2004) [4].

The theory of technology acceptance modelling (TAM) suggests that individuals' acceptance and use of new technologies are primarily influenced by perceived usefulness and perceived ease of use.TAM explains the subjective factors inherent in people's acceptance of Internet technologies (Davis, F., 1989; Magsamen-Conrad K. & Billotte-Verhoff C. & Dillon J., 2022) [5, 6]. And the theory of biased technological progress (BTA), which was developed by Western scholars on the basis of the theory of induced innovation and the theory of endogenous technological progress, reveals that the bias of technological progress leads to different proportional changes in the marginal rates of output of different factors of production, which causes changes in the rates of remuneration of different factors of production, and consequently changes in the shares of different factors received in the gross national income (Acemoglu Daron, 2002; 2003) [7, 8]. And it is also noted that the skill bias of technological progress will increase the demand for skilled labour, resulting in a sustained skill premium as wages of skilled labour continue to rise relative to wages of unskilled labour (Adachi H. & Inagaki K. & Nakamura T. & Osumi, Y., 2019; Meng F. & Wang W., 2021) [9, 10]. The above two theories become the theoretical basis for studying the impact of individual Internet use preferences on labour income. However, through literature analysis, it is found that academic research on the complex relationship between individuals' Internet use preferences and labour income is still insufficient, which provides space for this study.

In recent years, China, as a leader of developing countries, has an economic scale of more than 121 trillion yuan in 2022, ranking steadily as the second largest economy in the world, with per capita GDP reaching 85,698 yuan, exceeding the world's level of per capita GDP; moreover, the scale of the digital economy has reached 50.2 trillion yuan, with the total volume ranking steadily as the world's second, and the scale of Internet users has also reached 1,067 million people [11]. This provides valuable data for this study. Therefore, taking Chinese residents as the research object, this study aims to analyse the relationship between individual Internet usage preferences and labour income in the context of digital transformation, and tries to reveal the differences in the impact of different individuals' Internet usage preferences on their labour income. It is hoped that the results of this study can provide a basis for the government to formulate digital equity policies, and also remind residents to make better use of Internet resources to improve their human capital and obtain higher labour income.

## 2. Literature review

The development and application of computers have promoted the process of informatization of enterprises, realized the collection, storage, processing, transmission and utilization of information, changed and optimized the way enterprises manage and operate, and affected the labour market and workers' incomes. (1) It has realised automation and intelligence in the field of production. Through the mutual complementarity of computers and human skills, informatisation has enabled many tasks to be done automatically by computers and automated equipment, reducing the need for human resources and improving work efficiency and production quality. For example, automated robots on production lines can replace human beings in repetitive labour and improve productivity (Atrostic B. & Sang V. 2002; DeCanio, Stephen, 2016) [12, 13]; intelligent machines and equipment can carry out complex calculations and make decisions to improve the intelligence of work (Brynjolfsson E. & McAfee A., 2014; Francisco M. & Björn-Ola L., 2023) [14]. (2) It ensures information sharing and collaborative work in the enterprise. Through the application of information technology, instant information sharing and collaborative work can be achieved between different departments and teams. This leads to the optimisation and transformation of jobs, work management systems and organisational structures in firms (Cascio W. & Montealegre R., 2016) [15], and can also positively impact on product innovation, process improvement and employee skills (Ann B., Casey I. & Kathryn S., 2007) [16]. (3) The informatisation process has had a profound impact on the labour market. As the process of computerisation accelerates, low-skilled and low-educated workers in the labour market are more vulnerable to computerisation, while high-skilled and highly educated workers are more resilient to it (Frey C. & Osborne M. (2017) [17]. Also computerisation places higher skill requirements on workers (Acemoglu, D. & Autor, D. 2011) [18] and has led to significant declines in labour participation rates (Krueger, A. (2017) [19]; and automation (including computer use) has implications for the polarisation of jobs that will lead to a concentration of job growth in low-paying and high-paying positions, while at the same time will make the labour income gap more pronounced (Autor, D. and Salomons, A. (2018) [20]. It can be seen that acquiring the necessary IT skills becomes one of the most important conditions for finding a job (Florence Jaumotte, et al. 2023) [21]. (4) There is a significant correlation with wage income. For example, a study of data from the U.S. Census of Population showed that computer use can increase personal income by 10–15 per cent (Krueger A., 1993) [22]. A study using Australian census data found that computer use can increase personal income by 12–16 per cent (Miller P. & Mulvey C., 1997) [23]. And the results of the analysis of data from the level of enterprises in developed countries show that the wages of workers with skills in the use of computers increase by about 1% per year (Entorf H. &

Kramarz F., 1997) [24]. Whereas, data in China confirmed that the use of computers at work by workers in developing countries increased the return on workers' wages by 48.4% (Du Y., et al. 2023) [25]. And (Borghans L. & B. ter Weel2008) pointed out that different patterns of computer use also have a significant effect on the wage structure of workers [26].

As a new generation of Internet technologies such as big data, cloud computing, Internet of Things, artificial intelligence and other technologies continue to mature and accelerate their deep integration with various industries of the national economy, it accelerates the process of quality change, efficiency change, and power change in various industries, and has a more significant impact on people's production and life. (1) Promoting employment. On the one hand, the widespread use of the Internet has driven the increase in demand for zero-work economy jobs such as e-commerce platforms, online payment systems, logistics and distribution (Graham, M., Hjorth, I., & Lehdonvirta, V. 2017), and promoted equal employment opportunities for both urban and rural residents (Li G. & Qin J. 2022) [27, 28]. On the other hand, the use of Internet tools can significantly reduce the cost of job search for workers and increase the probability of job seekers obtaining a job, which in turn reduces the unemployment rate (Xin Jin, Baojie Ma & Haifeng Zhang, 2023) [29]; the use of the Internet by workers for skill enhancement and online social activities can significantly increase the accumulation of human capital and social capital, which in turn has a positive impact on the improvement of individual labour productivity and increase in income (Dettling L., 2017) [30]. In addition, Internet use also has a significant impact on individuals' employment decision-making behaviour (Mao Y., et al., 2019) [31]. (2) Enhancement of labour income. Not only can workers use the Internet at work to bring in additional income (Goss E. & Phillips J., 2002) [32], but using the Internet outside of work also has the potential to enable individuals to earn higher wages (DiMaggio P. & Bonikowski B., 2008) [33]. Clearly, Internet use can be an effective moderator of wage distortions for workers (Zhao X. et al., 2022) [34]. However, the income effect of Internet use varies significantly by workers' gender, marital status, age, level of education, and the content of Internet use (Philip L., 2017; Mao Y. et al., 2018) [35, 36]. Moreover, Internet technologies increase workers' job satisfaction and subjective well-being by improving access to information, creating new novel opportunities, facilitating social interactions, changing work patterns, and raising people's expectations of material wealth (Castellacci, F., & Viñas-Bardolet, C. 2019; Bloom, N. et al. 2015; Wu J. & Zhou C., 2023; Lohmann, S. 2015) [37–40]. In addition, the uncertainty of workers' income, labour rights, social security, and health risks in the context of Internet use are also of concern (Berg J., et al., 2018; Heeks R. 2017; Wood A. et al., 2019) [41–43].

The internet has brought about openness, reciprocity, sharing, and also "personalization." Individuals can engage in personalized internet activities such as searching, selecting, creating, and publishing based on their own interests and needs to meet their life and production requirements. However, due to differences among individuals in terms of nationality, gender, age, education level, family situation, and internet devices, there will inevitably be preferences in internet usage, leading to significant differences in the benefits gained from such usage (Castellacci F. & Tveito V., 2018) [44]. Therefore, this study utilizes microdata from China to specifically examine the relationship between internet usage preferences and labor income, and thereby determines the extent to which an individual's internet usage preferences affect their labor income. Based on this, the following hypotheses are proposed:

**Hypothesis 1:** Internet usage has a positive impact on improving individual labor income.

**Hypothesis 2:** The impact of internet usage preferences on individual labor income varies.

**Hypothesis 3:** There are significant regional differences in the impact of internet usage preferences on individual labor income.

**Hypothesis 4:** Enhancing residents' digital literacy has a positive impact on improving their
labor income.

## 3. Data sources, model specification, and variable description

### 3.1 Data sources

The China General Social Survey (CGSS) is the earliest nationwide, comprehensive and con-
tinuous academic survey of families in China. Since 2003, the survey team has conducted
annual questionnaire surveys on individuals in 125 counties (districts), 500 streets (townships
and towns), 1,000 neighbourhood (village) committees, and more than 10,000 households
across the country, systematically and comprehensively collecting data at multiple levels of
society, communities, households, and individuals in China, and summarising the trends of
social change in China. There were 12,582 valid samples in the 2017 survey data (CGSS2017),
containing 783 variables in modules such as the core module, social network, network society,
and household questionnaire [45]. These survey data on residents'(or households') personal
characteristics, income, frequency of Internet use, and Internet use preferences support the
empirical analyses of this study. In order to ensure the scientific nature of the data analysis, we
excluded the samples with missing key variables and obtained a total of 2072 sets of valid data.

### 3.2 Variables setting and description

Based on the topic and model design of this study, variables of interest, independent variables,
and control variables were selected from the CGSS2017 database, resulting in a total of 2013
sets, and descriptive statistics were conducted (see Table 1).

   **Dependent variable.**   Individual labor income, referred to as Income (INC). The answers
to the question "What was your personal labor income for the past year?" in the database were
used as the source data. Since the values of individual labour income obtained from the survey
are taken as integers and there is a significant spacing or separation between the values, i.e.,
INC is a discrete variable.

   **Independent variables.**   The individual's Internet usage preference, named **_Internet_**.
where the responses to the question "How often have you used the Internet in the past year?"
from the database were selected to measure the amount of time the individual spends using
the Internet, i.e., _Internet usage frequency (IUF)_. Select "In the past year, how often did you use
the Internet because of _Online social networking (OSN) / Online self-presentation (OSP) /
Online rights protection (ORP) / Online entertainment (OE) / Accessing the Internet" to measure
the amount of time an individual spent on the Internet. entertainment (OE) / Accessing online
information (AOI) / Online business (OB)_?". The questionnaire was designed to measure peo-
ple's preference or tendency to use the Internet. In addition, the responses to the question
"What percentage of the time do you use a computer for work during the week?" were selected
as a validation of the stability of the model. as an alternative variable to verify the robustness of
the model, i.e., proportion of computer usage in work (PCUW). Fig 1 shows the distribution
of different online activities at different frequencies (Never, Rarely, Sometimes, Often,
Always). The data shows that most people tend to engage in Online Self-Presentation (OSP)
either frequently or always, with the highest percentage of always (46.23%). Online entertain-
ment (OE) follows as an integral part of people's daily lives, with 44.45 per cent always. In a
similar vein, Online Social Networking (OSN) is also used frequently, with 42.04 per cent
always. Accessing online information (AOI) is also a common behaviour, always accessed at
41.67%. In contrast, Online Business (OB) and Online Rights Protection (ORP) were used less
frequently. While many people frequently or always conduct online business (OB), with 21.04

**Table 1. Descriptive statistics for variables.**

| Variable Categories | Variable Types | Variable Names | Obs. | Mean | Std. dev. | Min | Max | Instruction |
|---|---|---|---|---|---|---|---|---|
| Dependent variable | Income | INC | 2,072 | 45613.7 | 87990.7 | 0 | 2000000 | The currency is CNY |
| Independent variable | Internet usage frequency | IUF | 2,072 | 4.114 | 1.067 | 1 | 5 | There are 5 levels from low to high: 1, 2, 3, 4 and 5. |
| | Internet usage preferences | OSN | 2,072 | 3.695 | 1.072 | 1 | 5 | |
| | | OSP | 2,072 | 2.763 | 1.126 | 1 | 5 | |
| | | ORP | 2,072 | 1.899 | 0.977 | 1 | 5 | |
| | | OE | 2,072 | 3.337 | 1.112 | 1 | 5 | |
| | | AOI | 2,072 | 3.542 | 1.076 | 1 | 5 | |
| | | OB | 2,072 | 2.933 | 1.323 | 1 | 5 | |
| | Alternative Variables | PCUW | 2,072 | 18.276 | 28.979 | 0 | 100 | Values are percentages. |
| Control variable | Personal characteristics | Gender | 2,072 | 0.509 | 0.500 | 0 | 1 | 0 means male, 1 means female. |
| | | Age | 2,072 | 39.910 | 12.960 | 17 | 70 | Respondents' true age. |
| | | Age2 | 2,072 | 1760.7 | 1093.311 | 289 | 4900 | |
| | | HS | 2,072 | 3.816 | 0.958 | 1 | 5 | There are 5 levels of personal health, from the lowest to the highest, namely 1, 2, 3, 4 and 5. |
| | | EA | 2,072 | 6.689 | 3.247 | 1 | 13 | From "no education" to "master's degree and above," there are a total of 13 levels. |
| | Psychological characteristics | CA | 2,072 | 2.540 | 1.407 | 1 | 5 | There are 5 levels from lowest to highest: 1, 2, 3, 4 and 5. |
| | | DC | 2,072 | 3.905 | 0.953 | 1 | 5 | |
| | Social characteristics | SI | 2,072 | 2.775 | 0.995 | 1 | 5 | |
| | | SES | 2,072 | 2.337 | 0.858 | 1 | 5 | |
| | | RA | 2,072 | 0.774 | 0.419 | 0 | 1 | 0 means living in an urban area, 1 means living in a rural area. |
| | Personal Internet Use at Work | PIUW | 2,072 | 0.276 | 0.447 | 0 | 1 | 0 means the Internet is used at work, 1 means it is not. |

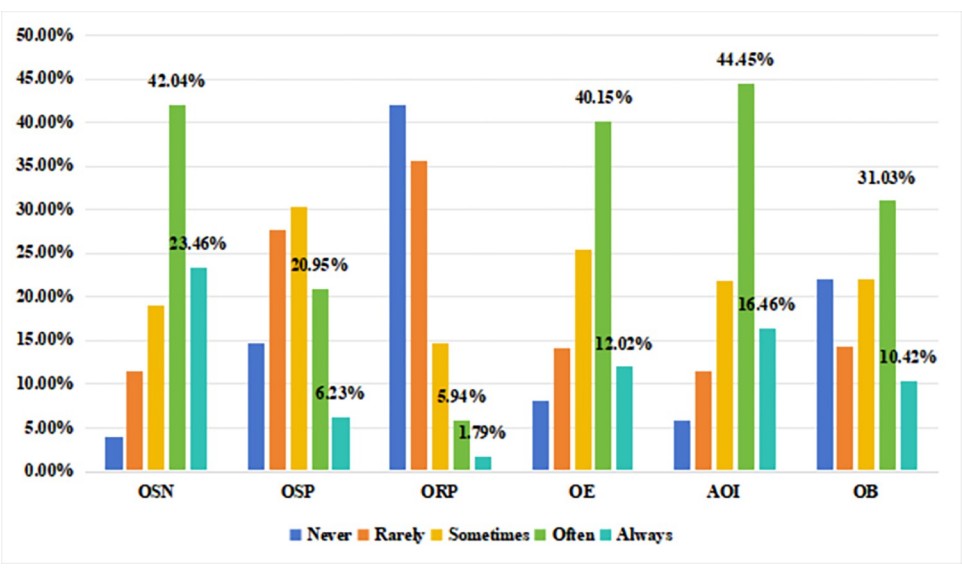

**Fig 1. Status of internet usage preference of survey respondents.**

per cent always doing so, a higher proportion (31.03 per cent) also never do so. Online Rights Protection (ORP), on the other hand, is the least frequently conducted activity with the highest percentage of never conducted (40.12%). This shows that online social networking, online self-presentation, online entertainment and accessing online information are the most frequently performed activities, while online rights protection and online business are relatively less frequently performed.

**Control variables.** Gender, Age, Age squared (Age2), Health status (HS), and Educational attainment (EA) were chosen as individual physiological characteristic variables. Cognitive abilities (CA) were measured using the answers to the question "In the past year, did you frequently read books / newspapers / magazines in your leisure time?" Psychological Depressive condition (DC) was measured by the frequency of feeling depressed or downhearted in the past four weeks. Additionally, social interaction (SI), socioeconomic status (SES), residential area (RA) were selected as social characteristic variables. In addition, "Does the company (or organization) you work for use the Internet to be more productive?" This response was selected as a measure of individual Internet use in the workplace, named "Personal Internet Use at Work" (PIUW).

## 3.3 Analyse of variable correlation

The results of the Pearson and Spearman correlation analysis in Table 2 show that each element in the matrix represents a correlation coefficient between two variables taking values ranging from -1 to +1. Among them, the independent variables IUF, OSN, OSP, ORP, OE, AOI, and OB are all significantly and positively correlated with the dependent variable INC, and the vast majority of the control variables also have significant correlation.

## 3.4 Model specification

The nature of people's use of the Internet is an important means of obtaining information, enhancing personal information capabilities and expanding the radius of employment. In order to estimate the effect of Internet usage preference on individual labour income, this paper constructs a functional Eq (1) for the effect of Internet usage preference on individual labour income by referring to Krueger's (1993) model setting form.

$$INC_i = \alpha + \beta_i Internet_i + \Sigma \gamma_i Z_i + \varepsilon_i \qquad (i = 1 \cdots n) \qquad (1)$$

Eq (2) can be obtained by placing the control variables containing physical, psychological, and social characteristics into Eq (1).

$$INC_i = \alpha + \beta_i Internet_i + \gamma_i Gender_i + \gamma_i Age_i + \gamma_i Age2_i + \gamma_i HS_i + \gamma_i EA_i \\ + \gamma_i CA_i + \gamma_i DC_i + \gamma_i SI_i + \gamma_i SES_i + \gamma_i RA_i + \gamma_i PIUW_i + \varepsilon_i \qquad (i = 1 \cdots n) \qquad (2)$$

In Eqs (1) and (2): $INC_i$ is the value of an individual's labour income; Interneti is an individual's frequency of Internet use and preference, including: $IUF_i$, $OSN_i$, $OSP_i$, $ORP_i$, $OE_i$, $AOI_i$ and $OB_i$; $Z_i$ denotes an individual's physical characteristics ($Gdender_i$, $Age_i$, $HS_i$, $EA_i$), psychological characteristics ($CA_i$ and $DC_i$), social characteristics ($SI_i$, $SES_i$, $RA_i$), and control variables such as Internet use status at work ($PIUW_i$); $\beta_i$ is the coefficient to be estimated; $\gamma_i$ is the coefficient of the control variables; and $\varepsilon_i$ is the residual term.

Since 'Internet usage preference' is a latent variable, it combines the effects of the frequency of multiple tendencies (or directions) of Internet usage on an individual's labour income. Suppose that the latent variable $L_i$ represents 'Internet usage preference', which is the result of the combined impact of different aspects of Internet usage (OSN, OSP, ORP, OE, AOI, OB). This latent variable can be viewed as the overall impact of Internet use on individual capabilities

**Table 2. Variable correlation analysis.**

| Variable | INC | IUF | OSN | OSP | ORP | OE | AOI | OB | Gender | Age | Age2 | HS | EA | CA | DC | SI | SES | RA | PIUW |
|---|---|---|---|---|---|---|---|---|---|---|---|---|---|---|---|---|---|---|---|
| INC | 1 | 0.206*** | 0.127*** | 0.090*** | 0.115*** | 0.083*** | 0.224*** | 0.300*** | 0.240*** | -0.061*** | -0.061*** | 0.150*** | 0.352*** | 0.137*** | 0.116*** | 0.039* | 0.191*** | 0.218*** | 0.405*** |
| IUF | 0.153*** | 1 | 0.394*** | 0.291*** | 0.207*** | 0.379*** | 0.403*** | 0.436*** | 0.060*** | -0.319*** | -0.319*** | 0.166*** | 0.383*** | 0.149*** | 0.064*** | 0.062*** | 0.054** | 0.194*** | 0.216*** |
| OSN | 0.110*** | 0.416*** | 1 | 0.428*** | 0.173*** | 0.451*** | 0.467*** | 0.501*** | -0.034 | -0.246*** | -0.246*** | 0.132*** | 0.279*** | 0.098*** | 0.054** | 0.108*** | 0.069*** | 0.112*** | 0.156*** |
| OSP | 0.065*** | 0.309*** | 0.467*** | 1 | 0.439*** | 0.360*** | 0.320*** | 0.448*** | -0.102*** | -0.307*** | -0.307*** | 0.156*** | 0.225*** | 0.070*** | 0.026 | 0.100*** | 0.104*** | 0.068*** | 0.117*** |
| ORP | 0.056** | 0.204*** | 0.200*** | 0.451*** | 1 | 0.231*** | 0.219*** | 0.377*** | 0.008 | -0.320*** | -0.320*** | 0.125*** | 0.252*** | 0.138*** | -0.023 | 0.061*** | 0.083*** | 0.073*** | 0.162*** |
| OE | 0.074*** | 0.389*** | 0.442*** | 0.380*** | 0.265*** | 1 | 0.517*** | 0.434*** | 0.029 | -0.297*** | -0.297*** | 0.143*** | 0.263*** | 0.119*** | 0.058*** | 0.087*** | 0.027 | 0.110*** | 0.093*** |
| AOI | 0.170*** | 0.422*** | 0.465*** | 0.351*** | 0.253*** | 0.517*** | 1 | 0.493*** | 0.110*** | -0.140*** | -0.140*** | 0.129*** | 0.398*** | 0.271*** | 0.104*** | 0.061*** | 0.102*** | 0.182*** | 0.207*** |
| OB | 0.216*** | 0.437*** | 0.500*** | 0.456*** | 0.378*** | 0.439*** | 0.494*** | 1 | 0.001 | -0.495*** | -0.495*** | 0.241*** | 0.448*** | 0.171*** | 0.042* | 0.072*** | 0.113*** | 0.197*** | 0.297*** |
| Gender | 0.135*** | 0.047** | -0.040* | -0.097*** | 0.013 | 0.024 | 0.108*** | 0.003 | 1 | 0.028 | 0.028 | 0.056** | 0.07*** | 0.046** | 0.065*** | -0.026 | -0.046** | -0.017 | 0.055** |
| Age | -0.038* | -0.306*** | -0.246*** | -0.305*** | -0.293*** | -0.300*** | -0.160*** | -0.494*** | 0.031 | 1 | 1.000*** | -0.304*** | -0.285*** | 0.021 | 0.033 | -0.02 | 0.053** | 0.039* | -0.152*** |
| Age2 | -0.061*** | -0.292*** | -0.233*** | -0.293*** | -0.277*** | -0.282*** | -0.152*** | -0.484*** | 0.038* | 0.987*** | 1 | -0.304*** | -0.285*** | 0.021 | 0.033 | -0.02 | 0.053** | 0.039* | -0.152*** |
| HS | 0.097*** | 0.163*** | 0.138*** | 0.166*** | 0.125*** | 0.154*** | 0.173*** | 0.248*** | 0.059*** | -0.298*** | -0.291*** | 1 | 0.189*** | 0.047** | 0.240*** | 0.091*** | 0.150*** | 0.047** | 0.142*** |
| EA | 0.285*** | 0.362*** | 0.284*** | 0.216*** | 0.218*** | 0.256*** | 0.386*** | 0.439*** | 0.069*** | -0.276*** | -0.256*** | 0.198*** | 1 | 0.394*** | 0.079*** | 0.001 | 0.163*** | 0.343*** | 0.356*** |
| CA | 0.125*** | 0.154*** | 0.093*** | 0.059*** | 0.102*** | 0.109*** | 0.258*** | 0.146*** | 0.047** | 0.051** | 0.076*** | 0.056** | 0.357*** | 1 | 0.105*** | 0.075*** | 0.112*** | 0.238*** | 0.177*** |
| DC | 0.071*** | 0.054** | 0.044** | 0.037* | -0.026 | 0.052** | 0.099*** | 0.046** | 0.055** | 0.025 | 0.032 | 0.268*** | 0.068*** | 0.110*** | 1 | 0.039* | 0.121*** | 0.032 | 0.043** |
| SI | 0.028 | 0.083*** | 0.118*** | 0.101*** | 0.062*** | 0.082*** | 0.059*** | 0.069*** | -0.025 | -0.02 | -0.024 | 0.100*** | -0.002 | 0.068*** | 0.045** | 1 | 0.087*** | -0.063*** | 0.026 |
| SES | 0.196*** | 0.067*** | 0.085*** | 0.105*** | 0.069*** | 0.047** | 0.122*** | 0.123*** | -0.041* | 0.058*** | 0.057*** | 0.181*** | 0.171*** | 0.110*** | 0.143*** | 0.087*** | 1 | 0.054** | 0.112*** |
| RA | 0.145*** | 0.191*** | 0.110*** | 0.066*** | 0.048** | 0.112*** | 0.180*** | 0.194*** | -0.017 | 0.051** | 0.067*** | 0.056** | 0.317*** | 0.223*** | 0.031 | -0.075*** | 0.053** | 1 | 0.176*** |
| PIUW | 0.269*** | 0.204*** | 0.160*** | 0.116*** | 0.138*** | 0.103*** | 0.209*** | 0.296*** | 0.055** | -0.160*** | -0.183*** | 0.153*** | 0.364*** | 0.164*** | 0.046** | 0.022 | 0.111*** | 0.176*** | 1 |

Lower-triangular cells report Pearson's correlation coefficients, upper-triangular cells are Spearman's rank correlation

* $p < 0.05$

** $p < 0.01$

*** $p < 0.001$.

and productivity. Therefore, Eq (3) can be derived.

$$INC_i = \theta_0 + \theta_1 L_i + \theta_2 Gender_i + \theta_3 Age_i + \theta_4 Age2_i + \theta_5 HS_i + \theta_6 EA_i + \theta_7 CA_i$$
$$+ \theta_8 DC_i + \theta_9 SI_i + \theta_{10} SES_i + \theta_{11} RA_i + \theta_{12} PIUW_i + \varepsilon_i \qquad (i = 1 \cdots n) \qquad (3)$$

where $L_i$ is determined by Eq (4).

$$L_i = \delta_0 + \delta_1 OSN_i + \delta_2 OSP_i + \delta_3 ORP_i + \delta_4 OE_i + \delta_5 AOI_i + \delta_6 OB_i + \varphi_i \quad (i = 1 \cdots n) \quad (4)$$

Assuming that the error terms and both follow normal distributions, the marginal probability density function of $INC_i$ given the other observed variables ($Z_i$) depends on the joint distribution of $INC_i$ and $L_i$. Since $L_i$ is unobserved, we need to integrate it to obtain the marginal likelihood function (5) for the observed data:

$$p(INC_i|Z_i) = \int p(INC_i, L_i|Z_i) dL_i \qquad (i = 1 \cdots n) \qquad (5)$$

## 4. Empirical analysis

In order to verify the impact of Internet Usage Frequency (IUF) and Internet Usage Preferences (OSN, OSP, ORP, OE, AOI, OB) on Individual Labour Income (INC) to verify the correlation and the degree of influence between them. For this purpose, the OLS model, the Zero-inflated Negative Binomial model and the Bayesian ZINB model were used for analysis and intercomparison, and the Conditional mixed-process model (CMP) was used for endogenous instrumental variables according to the characteristics of the data.

### 4.1 Benchmark regression analysis

To understand the effects and contributions of core variables in order to better assess the influence of other variables, this study employed a stepwise approach in performing Ordinary Least Squares (OLS) benchmark regression analysis with the inclusion of control variables. We specifically observed the changes in the p-values and regression coefficients of internet usage frequency and six usage preferences. The results in Table 3 indicate that prior to the inclusion of control variables, all independent variables exhibited statistical significance (P-value < 0.05). Following the OLS regression analysis on the entire sample, IUF, AOI, and OB still displayed statistical significance with Marginal utility on individual labor income of 3810.6, 2544.5, and 7483.2, respectively. This preliminary evidence suggests that hypotheses 1 and 2 put forth by the author are valid.

### 4.2 Zero-inflated Negative Binomial regression

The dependent variable 'personal labour income' has a large number of counts with zero values (22.88 per cent of zero values), and these data show a large degree of dispersion (see Fig 2). Firstly, a regression analysis was carried out using the Zero-inflated Poisson (ZIP) model, the results of which showed that the model was not applicable. Finally, the Zero-inflated Negative Binomial (ZINB) regression model was chosen to analyse the variables of interest and its results are shown in Table 4.

From Table 4, it can be seen that Model 8 shows a significant positive impact of IUF (P-value = 0.000, coefficient = 0.097) on personal labor income, indicating that higher internet usage frequency increases the likelihood of obtaining high income. The results of Models 9 to 14 demonstrate significant differences in the effects of the six internet usage preferences on personal labor income. OSN (P-value = 0.000, coefficient = 0.072) has a significant positive influence on labor income, which may be because individuals' use of the internet for social interaction helps in accumulating social capital and thus attaining higher labor income. OE (P-value = 0.005, coefficient = 0.051) also has a significant positive impact on labor income,

**Table 3. OLS benchmark regression results.**

| Variable | Model (1) | Model (2) | Model (3) | Model (4) | Model (5) | Model (6) | Model (7) |
|---|---|---|---|---|---|---|---|
| | Internet usage frequency(IUF) | Online social networking (OSN) | Online self-presentation (OSP) | Online rights protection (ORP) | Online entertainment (OE) | Accessing online information (AOI) | Online business (OB) |
| *Income(INC)* | 3810.6*** | 2712.10 | 951.20 | (422.40) | 792.50 | 2544.5* | 7483.2*** |
| | (3.58) | (1.83) | (0.43) | (-0.21) | (0.46) | (2.26) | (5.24) |
| Gender | 21971.3*** | 22560.2*** | 22482.2*** | 22283.0*** | 22240.9*** | 21827.9*** | 22336.4*** |
| | (6.39) | (6.51) | (6.26) | (6.43) | (6.45) | (6.38) | (6.51) |
| Age | 6735.7*** | 6699.8*** | 6637.9*** | 6577.4*** | 6640.1*** | 6593.2*** | 6865.0*** |
| | (9.64) | (9.63) | (9.59) | (9.49) | (9.26) | (9.59) | (9.82) |
| Age squared (Age2) | -78.21*** | -78.19*** | -77.68*** | -77.32*** | -77.76*** | -77.21*** | -76.77*** |
| | (-9.73) | (-9.72) | (-9.68) | (-9.71) | (-9.50) | (-9.69) | (-9.70) |
| Health status (HS) | 147.60 | 146.10 | 145.20 | 198.40 | 172.40 | 45.41 | (186.20) |
| | (0.09) | (0.09) | (0.08) | (0.12) | (0.10) | (0.03) | (-0.11) |
| Educational attainment (EA) | 5050.1*** | 5143.2*** | 5276.6*** | 5321.2*** | 5274.5*** | 5096.9*** | 4660.7*** |
| | (7.88) | (8.15) | (8.14) | (8.04) | (8.23) | (7.87) | (7.41) |
| Cognitive abilities (CA) | 962.80 | 1089.60 | 1075.10 | 1096.00 | 1054.90 | 823.00 | 844.50 |
| | (0.85) | (0.96) | (0.95) | (0.94) | (0.95) | (0.74) | (0.75) |
| Depressive condition (DC) | 2256.10 | 2304.10 | 2344.90 | 2335.90 | 2331.30 | 2241.80 | 2282.40 |
| | (1.25) | (1.28) | (1.31) | (1.33) | (1.30) | (1.25) | (1.27) |
| Social interaction (SI) | 1042.80 | 1047.50 | 1285.50 | 1396.50 | 1309.60 | 1262.40 | 858.70 |
| | (0.62) | (0.63) | (0.76) | (0.83) | (0.80) | (0.75) | (0.51) |
| Socioeconomic status (SES) | 14429.3*** | 14402.2*** | 14429.2*** | 14551.2*** | 14504.9*** | 14388.0*** | 13681.2*** |
| | (5.14) | (5.15) | (5.20) | (5.13) | (5.16) | (5.13) | (5.01) |
| Residential area (RA) | 12314.8*** | 13135.9*** | 13393.9*** | 13490.3*** | 13329.1*** | 13055.8*** | 10671.8*** |
| | (5.63) | (5.97) | (6.04) | (-6.14) | (5.96) | (5.95) | (4.90) |
| Personal Internet Use at Work(PIUW) | 26841.7*** | 26999.8*** | 27305.1*** | 27413.1*** | 27376.6*** | 27065.8*** | 24954.4*** |
| | (5.41) | (5.50) | (5.55) | (5.37) | (5.48) | (5.48) | (5.20) |
| N | 2072 | 2072 | 2072 | 2072 | 2072 | 2072 | 2072 |

t statistics in parentheses

* p<0.05

** p<0.01

*** p<0.001

potentially due to individuals using the internet for entertainment activities to alleviate work and life stress, resulting in increased labor income. AOI (P-value = 0.000, coefficient = 0.074) has a significant positive effect on labor income, likely because the internet lowers the threshold for information collection and provides a wealth of information resources, benefiting individuals who are adept at utilizing the internet to acquire information, thus potentially obtaining higher labor income. OB (P-value = 0.000, coefficient = 0.160) has a significant positive impact on personal labor income, possibly because online transactions contribute to expanding the scope of transactions, enhancing information transparency and convenience, as well as reducing transaction costs, all of which help increase personal labor income. This further confirms the correctness of hypotheses 1 and 2.

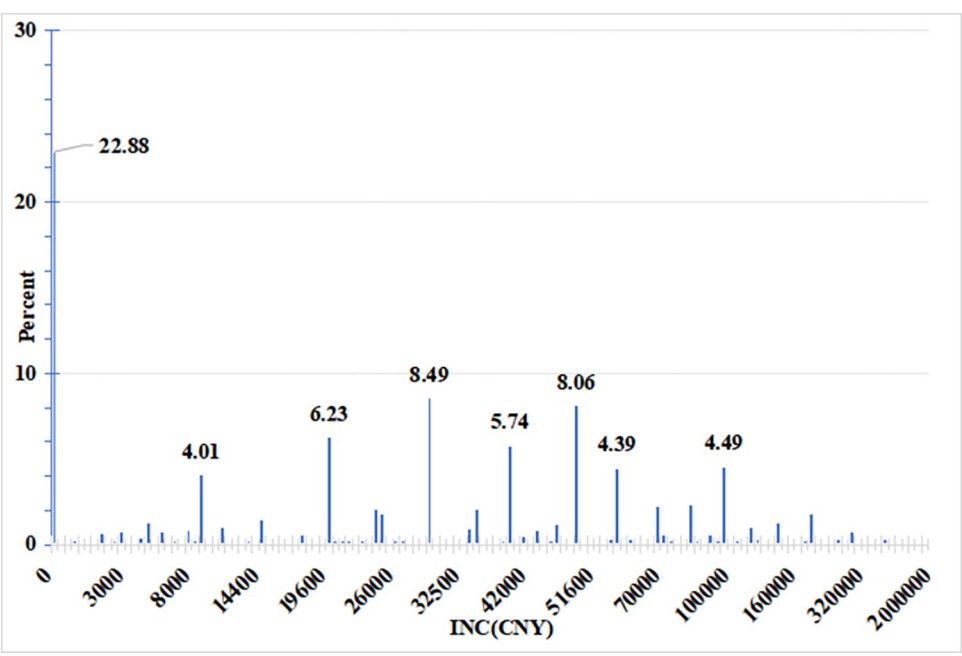

**Fig 2. Data distribution of the dependent variable "Personal annual labour income".**

The coefficients of the control variables in models (8) to (14) have the following characteristics: From the perspective of personal characteristics, both the internet usage frequency and the six internet usage preferences have a significant positive impact on labor income for individuals of different genders, indicating that internet usage preferences can enable males to obtain higher labor income. The coefficient of age is positively significant, and the coefficient of age squared is negatively significant, suggesting that labor income of workers will exhibit a inverted U-shaped trend with an initial increase followed by a decrease as age increases. The coefficient of health condition is positive, indicating that a better health condition of individuals leads to a higher likelihood of increasing labor income. The coefficient of education level is positive, indicating that individuals with higher educational attainment have a greater marginal effect on increasing labor income. From the perspective of psychological characteristics, the cognitive abilities accumulated from reading quantity within a year will have a positive impact on increasing labor income. From the perspective of social characteristics, the coefficient of individuals' social-economic class is positive, indicating that individuals' perception of their own social class has a positive effect on labor income. There exist significant differences in the impact of internet usage frequency and preferences on labor income between individuals living in cities and rural areas, with individuals living in cities having more opportunities to increase labor income compared to those living in rural areas. Moreover, an individual's income is likely to rise if the company (or organization) they work for uses the Internet to improve efficiency. These findings confirm the validity of the hypothesis 3 proposed by the author.

Furthermore, from Table 5, it can be observed that for every 1 unit increase in IUF, the likelihood of labor income increasing rises by approximately 11%, with a marginal effect of income increase of 4052.5. For every 1 unit increase in OSN, the likelihood of labor income increasing rises by approximately 7%, with a marginal effect of income increase of 2997.49. For every 1 unit increase in OE, the likelihood of labor income increasing rises by approximately 5%, with a marginal effect of income increase of approximately 2141.69. For every 1

**Table 4. Results of ZINB regression.**

| Variable | Model (8) | Model (9) | Model (10) | Model (11) | Model (12) | Model (13) | Model (14) |
|---|---|---|---|---|---|---|---|
| | Internet usage frequency(IUF) | Online social networking (OSN) | Online self-presentation (OSP) | Online rights protection (ORP) | Online entertainment (OE) | Accessing online information (AOI) | Online business (OB) |
| *Income(INC)* | 0.0974*** | 0.0718*** | 0.0248 | 0.0283 | 0.0513** | 0.0736*** | 0.160*** |
| | (4.97) | (3.89) | (1.37) | (1.43) | (2.80) | (3.67) | (9.16) |
| Gender | 0.377*** | 0.391*** | 0.389*** | 0.383*** | 0.378*** | 0.365*** | 0.377*** |
| | (10.06) | (10.37) | (10.23) | (10.15) | (10.02) | (9.63) | (10.20) |
| Age | 0.0608*** | 0.0608*** | 0.0597*** | 0.0599*** | 0.0605*** | 0.0580*** | 0.0639*** |
| | (5.81) | (5.79) | (5.66) | (5.67) | (5.75) | (5.54) | (6.19) |
| Age squared (Age2) | -0.0007*** | -0.0007*** | -0.0007*** | -0.0007*** | -0.0007*** | -0.0007*** | -0.0007*** |
| | (-5.56) | (-5.63) | (-5.57) | (-5.58) | (-5.59) | (-5.40) | (-5.45) |
| Health status (HS) | 0.0613** | 0.0593** | 0.0576** | 0.0574** | 0.0600** | 0.0584** | 0.0541* |
| | (2.79) | (2.69) | (2.60) | (2.59) | (2.71) | (2.66) | (2.51) |
| Educational attainment (EA) | 0.0766*** | 0.0791*** | 0.0829*** | 0.0824*** | 0.0809*** | 0.0774*** | 0.0704*** |
| | (10.65) | (11.07) | (11.71) | (11.60) | (11.38) | (10.65) | (9.90) |
| Cognitive abilities (CA) | 0.03 | 0.0330* | 0.03 | 0.0312* | 0.03 | 0.02 | 0.03 |
| | (1.93) | (2.12) | (1.96) | (2.00) | (1.88) | (1.50) | (1.74) |
| Depressive condition (DC) | 0.03 | 0.03 | 0.03 | 0.04 | 0.04 | 0.03 | 0.04 |
| | (1.61) | (1.70) | (1.68) | (1.87) | (1.75) | (1.57) | (1.85) |
| Social interaction (SI) | 0.01 | 0.01 | 0.01 | 0.02 | 0.01 | 0.01 | 0.01 |
| | (0.49) | (0.36) | (0.77) | (0.78) | (0.58) | (0.74) | (0.28) |
| Socioeconomic status (SES) | 0.246*** | 0.247*** | 0.248*** | 0.248*** | 0.248*** | 0.245*** | 0.228*** |
| | (11.04) | (11.03) | (11.03) | (11.06) | (11.06) | (10.99) | (10.28) |
| Residential area (RA) | 0.466*** | 0.486*** | 0.490*** | 0.492*** | 0.490*** | 0.482*** | 0.447*** |
| | (9.55) | (10.03) | (10.07) | (10.11) | (10.07) | (9.91) | (9.31) |
| Personal Internet Use at Work(PIUW) | 0.243*** | 0.246*** | 0.256*** | 0.258*** | 0.262*** | 0.250*** | 0.203*** |
| | (5.57) | (5.62) | (5.86) | (5.90) | (5.99) | (5.71) | (4.68) |
| inflate _cons | -1.215301 | | | | | | |

t statistics in parentheses

* p<0.05

** p<0.01

*** p<0.001

unit increase in AOI, the likelihood of labor income increasing rises by approximately 8%, with a marginal effect of income increase of approximately 3066.58. For every 1 unit increase in OB, the likelihood of labor income increasing rises by approximately 17%, with a marginal effect of income increase of approximately 6667.96. Comparing the Incident Rate Ratio (IRR) and Marginal utility of the internet usage preferences on labor income, the order is OB > AOI > OSN > OE.

## 4.3 Bayesian ZINB analysis

The use of Bayes prefixes for ZINB model fitting implies a Bayesian approach to estimating the model parameters and the ability to specify the prior distribution as well as use MCMC to

**Table 5. Incident Rate Ratio and marginal effects of internet usage preferences.**

| Variable | IRR | *Marginal utility* | *P-value* |
|---|---|---|---|
| **Internet usage frequency (IUF)** | 1.11 | 4052.5*** | 0.000 |
| **Online social networking (OSN)** | 1.07 | 2997.49*** | 0.000 |
| **Online self-presentation (OSP)** | 1.03 | 1034.66 | 0.172 |
| **Online rights protection (ORP)** | 1.03 | 1181.29 | 0.155 |
| **Online entertainment (OE)** | 1.05 | 2141.69** | 0.005 |
| **Accessing online information (AOI)** | 1.08 | 3066.58*** | 0.000 |
| **Online business (OB)** | 1.17 | 6667.96*** | 0.000 |

* p<0.05

**p<0.01

*** p<0.001

sample the posterior distribution. This approach provides more flexible modelling options and more accurate parameter estimates. The prior distributions of the model parameters in this paper are set by default, i.e., a standard or generic prior distribution is used without additional information. The seven images in Fig 3 (the vertical axis shows the frequency and the horizontal axis shows the range of values) show that after viewing the posterior distribution of intercepts it was found that IUF, OSN, ORP, OE, AOI, and OB significantly exclude zero (i.e., 0 is not within the 95% confidence interval), which indicates that the values in Table 6 are significant and plausible. Moreover, comparing the values of inflate _cons and IRR in Tables 3–5 almost coincide, which indicates that the ZINB model and its regression results are highly plausible.

## 4.4 Heterogeneity analysis

In order to further investigate the heterogeneity of the impact of internet usage preferences on labor income for different groups, this study analyzed the heterogeneity of Marginal utility based on the educational levels and regional factors of the survey participants. Due to the EA being divided into three categories: primary education (including illiterate, elementary school, junior high school, vocational high school level), secondary education (including high school, technical school, and specialized education level), and higher education (associate degree and above), there are a total of 13 educational levels. The regression results show that the Marginal utility of these three educational categories indicate that higher educational levels correspond to a larger effect of internet usage preferences on labor income (Table 7 and Fig 4). In addition, Fig 4 shows that the marginal utility of IUF, OSN, AOI, and OB for the group with higher education is significantly higher than that of the group with secondary and primary education, i.e., the higher the level of education the higher the possibility of utilizing the Internet to obtain high returns.

Due to China's vast territory, inter-regional natural resources, climate, etc. have caused significant differences in natural resource endowment among regions; and the strategy of prioritising the development of heavy industries and key regions implemented in the previous period has resulted in a highly unbalanced distribution of industrial structure, production factors, transport infrastructure, educational resources, digital infrastructure, etc. in different regions, which in turn has led to significant differences in the level of economic development and the income of the population (He S. & Chen J., 2023; Sun W. & Liu Y., 2023 [46, 47]). Therefore, this paper draws on Zhang Z. & Jiang Y. (2023) and Liu F. & Song R. (2023) to divide the 31 provinces in mainland China where the survey respondents are located into

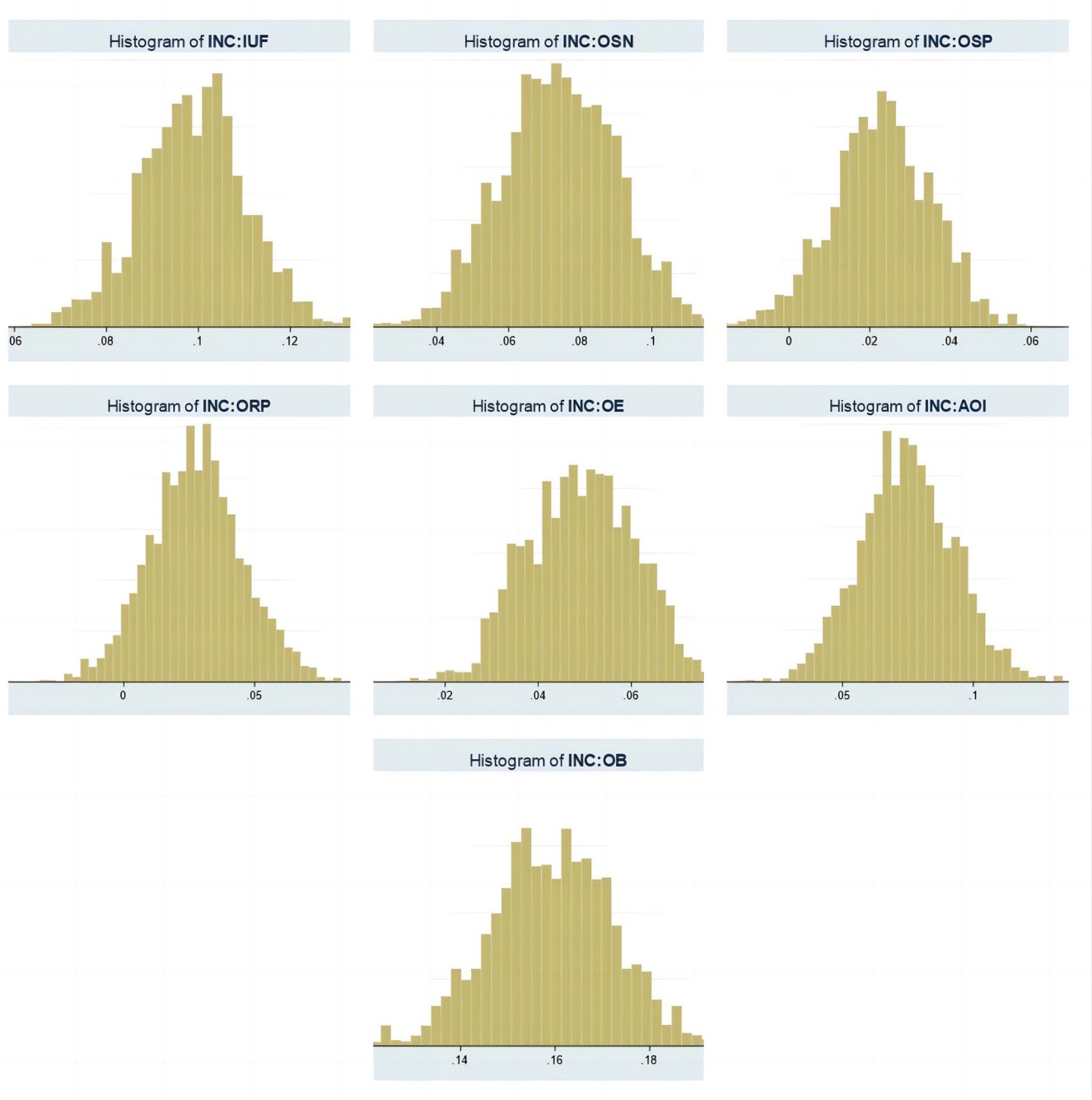

**Fig 3. Intercept posterior distribution of Internet usage preferences.**

three main regions, namely, the eastern, central, and western regions [48, 49], and conducts regression analyses. As shown in Table 8 and Fig 5: (1) In the eastern region IUF, AOI and OB show statistical significance, which suggests that people in the eastern region prefer to use the Internet to realize economic value and are more likely to obtain relatively higher marginal utility than in the central and western regions. (2) In the central region OSN, OSP, ORP, OE and OB show statistical significance and relatively higher marginal utility. And the diversified preferences of Internet use of people in the central region can make them gain in several aspects.

**Table 6. ZINB model fitting results using Bayes prefixes.**

| Variable | Mean | Std. dev. | Equal-tailed [95% cred. interval] | inflate_cons | IRR |
|---|---|---|---|---|---|
| Internet usage frequency (IUF) | 0.1054 | 0.0168 | 0.0704 0.1372 | -1.2152 | 1.1114 |
| Online social networking (OSN) | 0.0658 | 0.0158 | 0.0363 0.0971 | -1.2142 | 1.0681 |
| Online self-presentation (OSP) | 0.0247 | 0.0147 | -0.0037 0.0547 | -1.2171 | 1.0261 |
| Online rights protection (ORP) | 0.0252 | 0.0165 | -0.0059 0.0566 | -1.2171 | 1.0257 |
| Online entertainment (OE) | 0.0490 | 0.0095 | 0.0314 0.0674 | -1.2164 | 1.0503 |
| Accessing online information (AOI) | 0.0731 | 0.0185 | 0.0365 0.1095 | -1.2158 | 1.0761 |
| Online business (OB) | 0.1559 | 0.0148 | 0.1260 0.1832 | -1.2176 | 1.1688 |

(3) In the western region only AOI and OB show statistical significance. This suggests that people in the western region also focus on using the Internet to realize economic value. In conclusion, the significant differences in the frequency of Internet use and preferences of residents among the three regions resulted in significant differences in their labor income. This also reaffirms the validity of Hypothesis 3 proposed by the authors.

## 4.5 Endogenous treatment

While the previous analysis indicates that the frequency and preference of internet usage indeed have a significant positive impact on individual labor income, it is also possible for various internet usage preferences to be mutually correlated and mutually influential in reality. Additionally, the level of education also has a significant impact on labor income (Mincer J., 1974) [50]. Therefore, it is necessary to select appropriate instrumental variables for endogeneity testing in order to reduce estimation bias and improve the credibility and reliability of the research.

The state of Internet infrastructure construction is an important precondition for individuals to use the Internet. Since the state of Internet infrastructure construction is mainly determined by external factors such as government policies and the level of regional economic development, and these factors are not directly related to personal income, they satisfy the exogenous condition. Therefore, this paper uses the 'Fibre optic length (FOL)' of long-distance fibre optic cable line lengths of provinces in 2016 from the China Statistical Yearbook and

**Table 7. Education heterogeneity.**

| Variable | Primary Education | | Secondary education | | Higher Education | |
|---|---|---|---|---|---|---|
| | Marginal utility | P-value | Marginal utility | P-value | Marginal utility | P-value |
| Internet usage frequency(IUF) | 1724.12** | 0.005 | 5479.40*** | 0.000 | 11562.53** | 0.004 |
| Online social networking (OSN) | 2203.59*** | 0.001 | 1217.11 | 0.353 | 8298.79* | 0.014 |
| Online self-presentation (OSP) | 1673.06* | 0.017 | -796.70 | 0.529 | -167.37 | 0.945 |
| Online rights protection (ORP) | 1628.52* | 0.034 | 671.53 | 0.616 | 932.19 | 0.726 |
| Online entertainment (OE) | 1064.10 | 0.092 | 2069.11 | 0.125 | 1939.98 | 0.495 |
| Accessing online information (AOI) | 1182.40 | 0.063 | 3529.55* | 0.022 | 11178.06** | 0.009 |
| Online business (OB) | 4226.30*** | 0.000 | 6475.12 *** | 0.000 | 11971.57*** | 0.000 |

* $p < 0.05$

**$p < 0.01$

*** $p < 0.001$

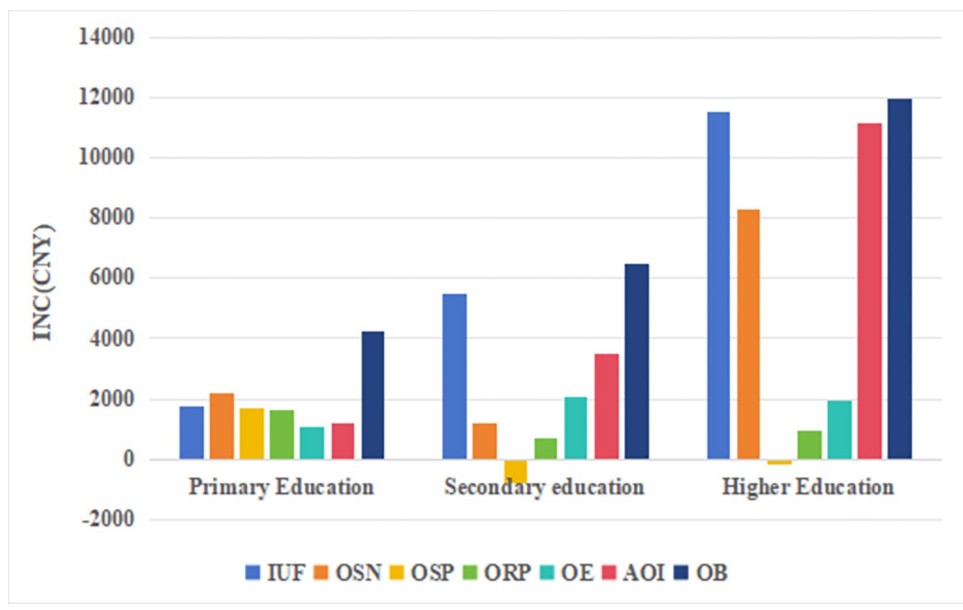

**Fig 4. Educational heterogeneity in the impact of internet use preferences on labour income.**

treats it as an instrumental variable for Internet usage preference to deal with endogeneity in the independent variables [51].

Based on the previous study, it is known that the dependent variable 'personal labour income' contains a large number of counting variables that take the value of zero, which makes it impossible to use the traditional two-stage least squares (2SLS) method. Therefore, this paper uses the conditional mixed process (CMP) model to solve the endogeneity problem (Roodman D., 2011) [52]. Using Stata17 statistical software, both FOL and FEA were put into the CMP model for regression analysis of the sample data. The results of the CMP regressions through the endogenous treatment in Table 7 show a significant and substantial increase in the positive marginal effects of IUF, OSN, OE, AOI, and OB, while OSP and ORP are similarly non-significant. These differences are perhaps caused by the fact that individual preferences

**Table 8. Regional heterogeneity.**

| Variable | Eastern Region | | Central Region | | Western Region | |
|---|---|---|---|---|---|---|
| | *Marginal utility* | *P-value* | *Marginal utility* | *P-value* | *Marginal utility* | *P-value* |
| **Internet usage frequency (IUF)** | 5989.87*** | 0.000 | 721.41 | 0.604 | 1103.92 | 0.210 |
| **Online social networking (OSN)** | 1826.29 | 0.123 | 3437.51* | 0.018 | 1128.22 | 0.209 |
| **Online self-presentation (OSP)** | 488.33 | 0.657 | 3626.67* | 0.014 | 463.46 | 0.651 |
| **Online rights protection (ORP)** | 125.55 | 0.921 | 4129.68** | 0.006 | 989.18 | 0.315 |
| **Online entertainment (OE)** | 833.53 | 0.466 | 3722.97* | 0.015 | 1564.12 | 0.069 |
| **Accessing online information (AOI)** | 3006.51* | 0.020 | 18.01 | 0.990 | 2821.42** | 0.003 |
| **Online business (OB)** | 6720.16*** | 0.000 | 5644.73 *** | 0.000 | 3558.65*** | 0.000 |

* p<0.05

**p<0.01

*** p<0.001

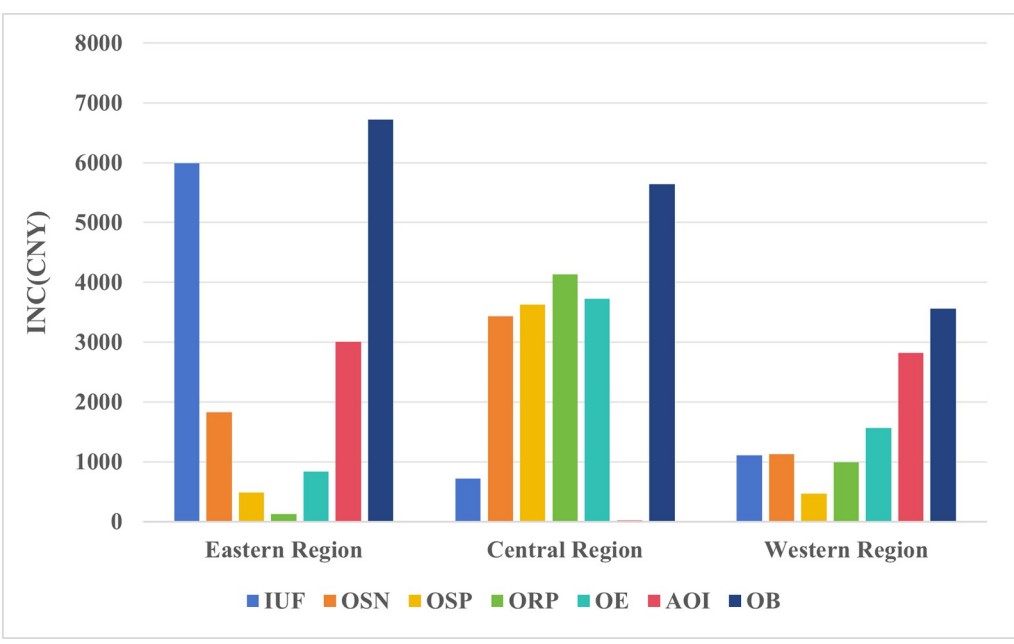

**Fig 5. Regional heterogeneity in the impact of internet use preferences on labour income.**

for OSN, OE, AOI and OB are more likely to result in higher individual labour income than OSP and ORP (Table 9).

In order to ensure the relevance of instrumental variables, the selected instrumental variable FOL was subjected to instrumental variable correlation test and exogeneity test, respectively. In Table 10, since the value of Prob > F of FOL with IUF, OSN, and OE is lower than 0.05 but F < 10, FOL and IUF, OSN, and OE are weakly correlated, i.e., weakly instrumental variables, and since the value of Prob > F of FOL with AOI, and OB is lower than 0.05 and F > 10, FOL is considered to be strongly correlated with AOI, and OB, i.e., weakly Instrumental variables. In Table 11, the P values of instrumental variable FOL with IUF, OSN, OSP, ORP, OE, AOI, and OB are significantly less than 0.05, i.e., FOL is significantly exogenous. These results

**Table 9. Endogeneity treatment results.**

| Variable | ZINB Regression | | CMP Regression | |
|---|---|---|---|---|
| | *Marginal utility* | *P-value* | *Marginal utility* | *P-value* |
| **Internet usage frequency (IUF)** | 4052.5*** | 0.000 | 105568.5 ** | 0.026 |
| **Online social networking (OSN)** | 2997.49*** | 0.000 | 111121.3 * | 0.011 |
| **Online self-presentation (OSP)** | 1034.66 | 0.172 | -2457.323 | 0.736 |
| **Online rights protection (ORP)** | 1181.29 | 0.155 | 6865.711 | 0.402 |
| **Online entertainment (OE)** | 2141.69** | 0.005 | 120457.1 * | 0.042 |
| **Accessing online information (AOI)** | 3066.58*** | 0.000 | 83824.77 ** | 0.008 |
| **Online business (OB)** | 6667.96*** | 0.000 | 86817.32** | 0.009 |
| **Proportion of computer usage in work (PCUW)** | 232.85*** | 0.000 | 4166.924 ** | 0.009 |

* $p < 0.05$

** $p < 0.01$

*** $p < 0.001$

Table 10.  Tests of correlation of instrumental variables.

| Variable | R-sq. | Adjusted R-sq. | Partial R-sq. | F(1,2059) | Prob > F | Instrumental Variables FOL |
|---|---|---|---|---|---|---|
| Internet usage frequency (IUF) | 0.2092 | 0.2046 | 0.0043 | 8.8370 | 0.003 | Weak Instrumental Variables |
| Online social networking (OSN) | 0.1393 | 0.1343 | 0.0035 | 7.2567 | 0.0071 | Weak Instrumental Variables |
| Online self-presentation (OSP) | 0.1429 | 0.1379 | 0.0001 | 0.2595 | 0.6105 | Correlation is not significant |
| Online rights protection (ORP) | 0.1243 | 0.1192 | 0.0003 | 0.5983 | 0.4393 | Correlation is not significant |
| Online entertainment (OE) | 0.1447 | 0.1398 | 0.0028 | 5.7651 | 0.0164 | Weak Instrumental Variables |
| Accessing online information (AOI) | 0.2016 | 0.197 | 0.0066 | 13.6611 | 0.0002 | Instrumental Variables |
| Online business (OB) | 0.386 | 0.3825 | 0.0053 | 10.9302 | 0.001 | Instrumental Variables |

indicate that the instrumental variable FOL is overall a valid instrumental variable despite its weak correlation with some of the variables.

## 4.6 Robustness analysis

To verify the robustness of the model, this study selected the variable "What percentage of your work in a typical week involves using the internet?" (PCUW) from the database as a replacement for the independent variable and conducted ZINB regression analysis and CMP regression. The results show that the higher the proportion of internet usage in work, the more significant the increase in labor income (Last row of Table 9). This indicates that the model set in this study is very robust.

Currently, in the face of the wave of digitisation, major countries around the globe are engaged in actions to enhance the digital literacy of their citizens in terms of their ability to access, understand, evaluate, create and disseminate information in the digital environment (Eshet-Alkalai Y., 2004) [53]. In practice, through systematic digital literacy education activities that include but are not limited to basic computer skills, information retrieval, information assessment, media use, cybersecurity awareness, digital tools, digital communication, digital creativity, and data literacy, people not only learn the necessary information processing skills to better use the Internet to access the knowledge and information they need, but also have a significant impact on crossing the digital divide and enhance equity in the benefits of Internet use (UIS., 2018; Ferrari A., 2019) [54, 55]. Therefore, in this paper, we use the data of Internet use preference multiplied with Educational attainment as well as Frequency of study in leisure

Table 11.  Tests of exogeneity of instrumental variables.

| Variable | Durbin (score) chi2(1) | Wu-Hausman F(1,2058) |
|---|---|---|
| Internet usage frequency(IUF) | 13.0403 (p = 0.0003) | 13.0403 (p = 0.0003) |
| Online social networking (OSN) | 13.0403 (p = 0.0003) | 13.0403 (p = 0.0003) |
| Online self-presentation (OSP) | 13.9857 (p = 0.0002) | 13.9856 (p = 0.0002) |
| Online rights protection (ORP) | 13.9701 (p = 0.0002) | 13.9699 (p = 0.0002) |
| Online entertainment (OE) | 13.8012 (p = 0.0002) | 13.7998 (p = 0.0002) |
| Accessing online information (AOI) | 13.2429 (p = 0.0003) | 13.2381 (p = 0.0003) |
| Online business (OB) | 11.8432 (p = 0.0006) | 11.8308 (p = 0.0006) |

**Table 12. Interaction regression.**

| Variable | ZINB Regression | | CMP Regression | |
|---|---|---|---|---|
| | *Marginal utility* | *P-value* | *Marginal utility* | *P-value* |
| Internet usage frequency–Improve Digital literacy(IUF –IDL) | 53.01*** | 0.019 | 323.10*** | 0.000 |
| Online social networking–Improve Digital literacy(OSN –IDL) | 49.12* | 0.035 | 350.58*** | 0.000 |
| Online self-presentation–Improve Digital literacy(OSP –IDL) | 16.73 | 0.488 | 313.06* | 0.013 |
| Online rights protection–Improve Digital literacy (ORP –IDL) | -2.20 | 0.939 | 301.88 | 0.079 |
| Online entertainment–Improve Digital literacy (OE–IDL) | 34.95 | 0.145 | 305.50** | 0.004 |
| Accessing online information–Improve Digital literacy (AOI–IDL) | 47.91*** | 0.048 | 369.71*** | 0.000 |
| Online business–Improve Digital literacy (OB–IDL) | 113.93*** | 0.000 | 428.58.96*** | 0.000 |

* p<0.05
**p<0.01
*** p<0.001

time to derive an interaction term of the three, named Improve Digital literacy (IDL). Using ZINB regressions and CMP regressions, the results show that enhancing the frequency of digital literacy education and self-study related to Internet technology has a significant contributing effect on improving IDL (Table 12). This again shows that the model set up by the authors is robust.

## 4.7 Analysis of the mediating effect mechanism of education

Since in reality education has a significant contribution to both Internet use preference and labour income [56, 57]. Therefore, the authors establish a mediation effect model with $INC_i$ as the dependent variable, Interneti as the independent variable, and $EA_i$ as the mediating variable. And the mediation effect regression analysis was conducted in STATA software using Bootstrap method.

$$INC_i = \rho_i Internet_i + \Sigma\gamma_i Z_i + \mu_{1i} \qquad (i = 1 \cdots n) \tag{6}$$

$$EA_i = \varphi_i Internet_i + \Sigma\gamma_i Z_i + \mu_{2i} \qquad (i = 1 \cdots n) \tag{7}$$

$$INC_i = \delta_i Internet_i + \theta_i EA_i + \Sigma\gamma_i Z_i + \mu_{3i} \qquad (i = 1 \cdots n) \tag{8}$$

In Eqs (6) (7) (8), $\rho_i$ is the total effect of the independent variable $Internet_i$ on the dependent variable $INC_i$; $\varphi_i$ is the effect of the independent variable Interneti on the mediating variable $EA_i$; $\theta_i$ is the effect of the mediating variable $EA_i$ on the dependent variable $INC_i$ after controlling for the effect of the independent variable $Internet_i$; $\delta_i$ is the effect of the direct effect of the independent variable $Internet_i$ on the dependent variable $INC_i$; $\Sigma\gamma_i Z_i$ is the control variable; $\mu_{1i}, \mu_{2i}, \mu_{3i}$, are the error terms.

Referring to scholars Wen Zhonglin et al.'s (2014) test step-by-step analysis method of causal stepwise regression method, it can be seen [58]: (1) The mediating variable EA presents a significant mediating effect on the independent variables IUF, AOI, OB and PCUW. (2) The mediating variable EA has a partial mediating effect on the independent variable OSN. (3) The mediating variable EA has a masking effect on the independent variables ORP, OSP and OE

**Table 13. Mediating effect analysis of the labour income effect of education level on individual internet use preferences.**

| Variable | Indirect effect | P-value | Direct effect | P-value | Total effect | P-value |
|---|---|---|---|---|---|---|
| Internet usage frequency(IUF) | 2505.70*** | 0.000 | 3810.61*** | 0.000 | 6316.31*** | 0.000 |
| Online social networking (OSN) | 2101.78*** | 0.000 | 2712.10 | 0.071 | 4813.88** | 0.002 |
| Online self-presentation (OSP) | 1162.64*** | 0.000 | 951.19 | 0.656 | 2113.83 | 0.333 |
| Online rights protection (ORP) | 1088.52** | 0.002 | -422.38 | 0.832 | 666.14 | 0.737 |
| Online entertainment (OE) | 1505.09*** | 0.000 | 792.46 | 0.635 | 2297.544 | 0.182 |
| Accessing online information (AOI) | 2964.01*** | 0.000 | 2544.53* | 0.028 | 5508.54*** | 0.000 |
| Online business (OB) | 2435.63*** | 0.000 | 7483.21*** | 0.000 | 9918.83*** | 0.000 |
| Proportion of computer usage in work (PCUW) | 104.11*** | 0.000 | 488.60*** | 0.000 | 592.71*** | 0.000 |

\* $p<0.05$

\*\*$p<0.01$

\*\*\* $p<0.001$

(Table 13). Therefore, improving education level is conducive to enhancing individual Internet use preference for labour income effect.

## 5. Discussion

Based on the previous empirical results, it can be seen that (1) regular use of the Internet helps to increase the level of Internet skills, which enhances the likelihood of earning a high income. Effective online social networking can accumulate social capital and thus enhance labour income. Moderate online entertainment can relieve work and life stress and enhance labour income. People who are good at accessing online information may earn higher labour income. People who regularly engage in online business have higher labour income. (2) However, individuals' physical characteristics, psychological characteristics, and social characteristics will lead to significant differences in the labour income effects of Internet use preferences. (3) There are large differences between urban and rural areas and regions in the effect of Internet usage preference on labour income, with the overall phenomenon of economically developed regions being higher than underdeveloped regions. (4) Increasing the level of digital literacy education can increase labour income. It can be seen that this study proves the significant effect of individual Internet use preference on their labour income, which also shows that the original hypothesis of this paper is correct.

Based on the Technology Acceptance Model (TAM), it can be seen that perceived usefulness refers to the value (or benefit) that an individual perceives in using a certain technology for his or her work or life, while perceived ease of use refers to the individual's perception of the ease or difficulty of using a certain technology. In the context of the digital transformation of the economy and society, most people choose to accept and bias the use of a certain Internet technology only after considering the actual value of that technology to their daily life and work and assessing the ease of mastering that technology. However, this process also exhibits more complex real-world outcomes due to significant differences in the physical, psychological, and social characteristics of individuals [59]. Therefore, in essence, the rapid innovation and popularisation of Internet technology (or digital technology) has become a trigger for people's diverse needs for Internet technology in the areas of life, leisure and entertainment, as well as work, etc.; and people's acceptance of a certain kind of Internet technology is like a catalyst for people's acceptance of that technology on the basis of their perception of the value of

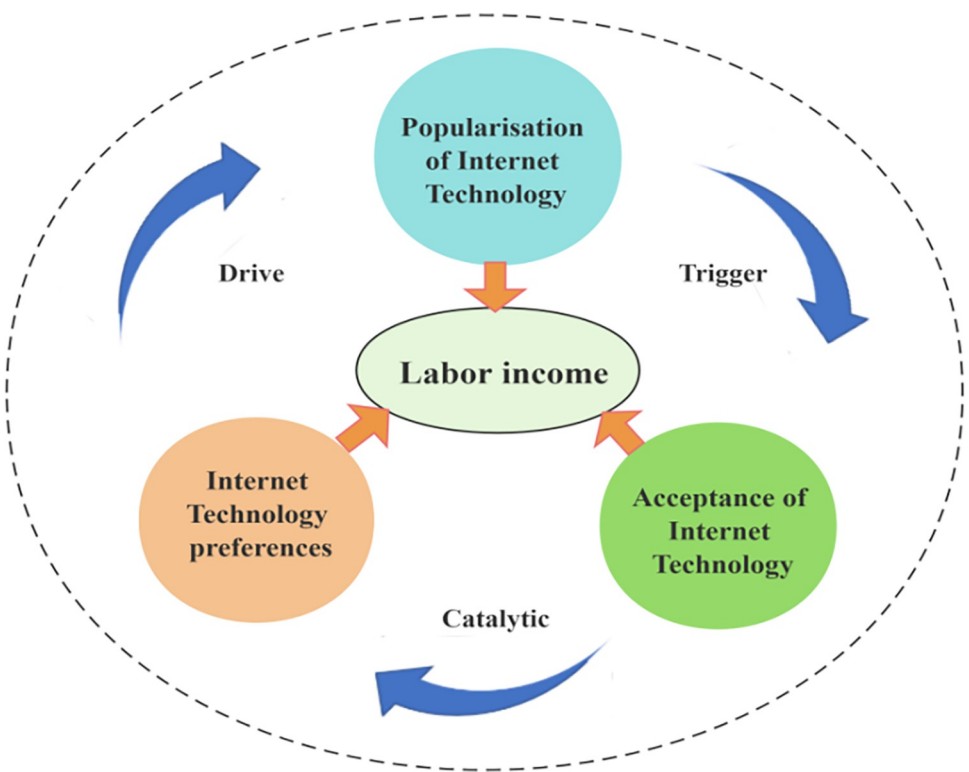

**Fig 6. Mechanisms of the impact of internet use preferences on individual labour income.**

the Internet technology and the difficulty of mastering it and assessing the difficulty of mastering it. In addition, according to the theory of biased technological progress, the current worldwide wave of digitisation has increased labour productivity, which has led to a significant increase in labour incomes for those who are digitally skilled (Acemoglu D., 2011; Sun, F., 2023) [60, 61]. In this context, the choice to learn or become proficient in the use of certain Internet technologies in order to gain a competitive advantage and higher value in the labour market has become an important driver for workers to improve their labour income. And this driving force will lead to further innovation and popularisation of Internet technologies, which will then trigger and catalyse the acceptance and selection of new technologies, and form an infinite cycle of significantly increasing individual labour income (Fig 6).

## 6. Conclusion

Overall, this study demonstrates that individuals' Internet use preferences have a positive effect on increasing individual labour income. It also reveals that this positive effect is highly heterogeneous depending on an individual's level of education and region of residence. It also demonstrates the importance of improving the digital literacy of the population in order to increase their labour income. However, due to the limitations of the CGSS2017 microdataset used in terms of the type of Internet usage preferences, sample size, and time, this study only demonstrates the positive effect of online social interaction, online entertainment, online information access, and online commerce on individual labour income. Therefore, in future research, on the one hand, we should follow the pace of digital technology innovation to further expand the types of Internet usage preferences, such as digital finance, artificial intelligence, digital creation, data analytics, etc.; on the other hand, we should work with our peers

to carry out more extensive surveys in both developed and developing countries, and to obtain a larger amount of survey data. Improvements in these two areas will not only provide a more realistic and comprehensive verification of the impact of Internet technology usage preferences on individual labour income and the labour market, but may also better explain the income inequality caused by differences in digital literacy.

Currently, the digital divide is widespread around the world. Significant differences in labour income caused by people's Internet usage preferences are likely to be an important factor in exacerbating the gap between rich and poor households. Therefore, the authors put forward the following suggestions: first, the government should increase the construction of Internet infrastructure and accelerate the process of informatisation in backward areas. Second, a segmented (youth, adult, elderly), graded (basic application, professional enhancement, innovation and creation) and categorised (general group, special group) approach should be adopted to establish a system for cultivating and evaluating the digital literacy of the whole population, so as to enhance the digital literacy of the whole population. Third, it is necessary to strengthen guidance on Internet use behaviour to prevent the negative impact of excessive entertainment and Internet addiction on employment and income. Fourth, a certain amount of digital literacy education subsidy should be provided to groups such as the unemployed, low-income earners, persons with disabilities, housewives and the elderly, so as to safeguard their basic livelihoods and promote their motivation to participate in digital literacy education and training. In addition, residents should make full use of the digital education resources provided by the Government to proactively improve their digital literacy levels, so as to lay the foundation for increased labour income.

## Acknowledgments

I would like to thank all the reviewers and editors for their valuable criticisms and revisions of this paper, as well as Professor Olga P. Nedospasova of the Faculty of Economics and Management, Tomsk State University, for inspiring and guiding the idea of writing this paper. I would also like to thank my wife, Xiaoxia Zhang, for taking good care of the family and for her assistance in writing this paper.

## Author Contributions

**Conceptualization:** Kefeng Yuan.

**Data curation:** Kefeng Yuan.

**Formal analysis:** Kefeng Yuan.

**Funding acquisition:** Kefeng Yuan.

**Investigation:** Xiaoxia Zhang.

**Methodology:** Kefeng Yuan, Xiaoxia Zhang.

**Project administration:** Xiaoxia Zhang.

**Supervision:** Xiaoxia Zhang.

**Validation:** Kefeng Yuan.

**Visualization:** Kefeng Yuan, Xiaoxia Zhang.

**Writing – original draft:** Kefeng Yuan, Xiaoxia Zhang.

**Writing – review & editing:** Kefeng Yuan, Xiaoxia Zhang.

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
