## [Decision Letter · Decision Letter 0]

25 Oct 2023

PONE-D-23-30708The Impact of Internet Usage Preferences on Labor Income : Evidence from ChinaPLOS ONE

Dear Dr. Yuan,

Thank you for submitting your manuscript to PLOS ONE. After careful consideration, we feel that it has merit but does not fully meet PLOS ONE’s publication criteria as it currently stands. Therefore, we invite you to submit a revised version of the manuscript that addresses the points raised during the review process.

We look forward to receiving your revised manuscript.

Kind regards,

Yasuko Kawahata

Academic Editor

PLOS ONE

Journal Requirements:

2. Thank you for stating the following financial disclosure: "This work was supported by grants from the National Social Science Foundation of China (No.22BJY045), Fujian Provincial Science and Technology Department (No. 2023R0071) and Ningde Normal University (No. 2022FZ01)."

3. Thank you for stating the following in the Acknowledgments Section of your manuscript: "This work was supported by grants from the National Social Science Foundation of China (No.22BJY045), Fujian Provincial Science and Technology Department (No. 2023R0071)  and Ningde Normal University (No. 2022FZ01)."

Please remove any funding-related text from the manuscript and let us know how you would like to update your Funding Statement. Currently, your Funding Statement reads as follows: "This work was supported by grants from the National Social Science Foundation of China (No.22BJY045), Fujian Provincial Science and Technology Department (No. 2023R0071)  and Ningde Normal University (No. 2022FZ01)."

Reviewers' comments:

Reviewer's Responses to Questions

**Comments to the Author**

1. Is the manuscript technically sound, and do the data support the conclusions?

Reviewer #1: Yes

Reviewer #2: Partly

Reviewer #3: Partly

Reviewer #4: No

2. Has the statistical analysis been performed appropriately and rigorously? 

Reviewer #1: Yes

Reviewer #2: No

Reviewer #3: Yes

Reviewer #4: No

3. Have the authors made all data underlying the findings in their manuscript fully available?

Reviewer #1: Yes

Reviewer #2: No

Reviewer #3: Yes

Reviewer #4: No

4. Is the manuscript presented in an intelligible fashion and written in standard English?

Reviewer #1: Yes

Reviewer #2: No

Reviewer #3: Yes

Reviewer #4: No

5. Review Comments to the Author

Reviewer #1: The proposed paper is devoted to the impcat of internet usage preferences on labor income. The material is focused on Chinese region and Chinese economy specifics. The processed paper is well designed and existing knowledgap is specified through the relevant literature overview. The objectives of the paper are also relevant and the proposed methodology is suitable and well explained. The results part of the paper is rational and individual results and findings are relevant and supported by relevant arguments. The final discussion part of the paper is not well processed - it should be separted from the conclussion and key findings related to already published outputs should be highlighted. The final conclussion is too brief. There is no relevant information related to paper's limitations. Also some recommendations for future research are missing.

Reviewer #2: This topic is in accordance with the context of 4.0 industrial revolution. The technology (Internet usage) has a significant contribution to the society and labor's income. Author(s) apply regression model (1) to estimate the internet usage on labor's income.

In terms of statistics and regression, this approach seems to be reasonable. However, the model does not clearly explain the mechanism of impact of technology (or internet use) on improving workers' income. Note that income must be linked to real economy activities. In equation 1, there is almost no mention of real economic factors. In other words, we need a model that explains the relationship between technology (the Internet) and income through the workings of the real economy.

Without theoretical background, the research resembles an econometrics exercise.

Reviewer #3: Using the data of China General Social Survey (CGSS) 2017, this paper uses Stata statistical software to build a model to analyze the impact of individual Internet use preference on labor income. It provides new evidence for understanding how Internet usage preferences affect labor income and highlights the importance of digital literacy, providing useful insights for policymakers and residents alike. I think the paper needs further revision.

1. The paper mentions the importance of digital literacy, and can further explain the specific content of digital literacy to help readers better understand it. In addition, the purpose, motivation and innovation of this paper should be further emphasized.

2. In Section 3, please explain the model and functional equations in detail so that the reader can better understand the research methods.

3. For functional equation (1), please explain the meaning of "" in the equation.

4. Improve the clarity of the bar chart in Figure 1.

5. Literature should be updated to add the latest research results of famous journals in the past three years.

Reviewer #4: The identification strategy needs to be revised to disentangle the effect of the internet on education and access to measure its effects on wages. The paper study an interesting topic but needs a better empirical strategy.

6. PLOS authors have the option to publish the peer review history of their article (what does this mean?). If published, this will include your full peer review and any attached files.

Reviewer #1: No

Reviewer #2: No

Reviewer #3: No

Reviewer #4: No

---

## [Author Response · Author response to Decision Letter 0]

8 Mar 2024

Response to Reviewers

Reviewer #1: The proposed paper is devoted to the impcat of internet usage preferences on labor income. The material is focused on Chinese region and Chinese economy specifics. The processed paper is well designed and existing knowledgap is specified through the relevant literature overview. The objectives of the paper are also relevant and the proposed methodology is suitable and well explained. The results part of the paper is rational and individual results and findings are relevant and supported by relevant arguments. The final discussion part of the paper is not well processed - it should be separted from the conclussion and key findings related to already published outputs should be highlighted. The final conclussion is too brief. There is no relevant information related to paper's limitations. Also some recommendations for future research are missing.

Author's Response:Thank you very much for your recognition of my article and your valuable modifications. I have the following modifications in response to your modifications, please criticise and correct me.

1. Revision of the discussion part of the paper: In the context of the popularisation of Internet technology, the formation mechanism of Internet usage preference is discussed based on the technology acceptance model and the theory of biased technological progress, and a logical relationship diagram is drawn to make it easier for readers to understand the mechanism.

2. Modification of the conclusion of the thesis: (1) The limitation of data and the limitation of variable selection are added. (2) Pointed out the future research direction and suggestions from two aspects.

Reviewer #2: This topic is in accordance with the context of 4.0 industrial revolution. The technology (Internet usage) has a significant contribution to the society and labor's income. Author(s) apply regression model (1) to estimate the internet usage on labor's income.

In terms of statistics and regression, this approach seems to be reasonable. However, the model does not clearly explain the mechanism of impact of technology (or internet use) on improving workers' income. Note that income must be linked to real economy activities. In equation 1, there is almost no mention of real economic factors. In other words, we need a model that explains the relationship between technology (the Internet) and income through the workings of the real economy.

Without theoretical background, the research resembles an econometrics exercise.

Author's Response:

Thank you very much for your valuable comments on my article. In response to your comments, I have made the following changes, and I would be grateful for your criticisms and corrections.

1. Income is not linked to the real economy: the survey question "Do you use the Internet at your workplace to improve your work efficiency?" was found in the CGSS2017 database and its data was used as a measure of an individual's Internet use at work. survey question and used its data as a measure of an individual's internet use at work, Personal Internet Use at Work (PIUW). The variable showed statistical significance from the test of ZINB and CMP endogeneity model.

2. Modification of the influence mechanism of the thesis: In the discussion part of the thesis, the formation mechanism of Internet usage preference is discussed on the basis of the technology acceptance model and the theory of biased technological progress, and a logical relationship diagram is drawn to make it easier for readers to understand the mechanism.

3. Modification of the conclusion of the thesis: (1) The limitation of data and the limitation of variable selection are added. (2) Pointed out the future research direction and suggestions from two aspects.

Reviewer #3: Using the data of China General Social Survey (CGSS) 2017, this paper uses Stata statistical software to build a model to analyze the impact of individual Internet use preference on labor income. It provides new evidence for understanding how Internet usage preferences affect labor income and highlights the importance of digital literacy, providing useful insights for policymakers and residents alike. I think the paper needs further revision.

Author's Response:

Thank you very much for your valuable comments on my article. In response to your comments, I have made the following changes, and I would be grateful for your criticisms and corrections.

1. The paper mentions the importance of digital literacy, and can further explain the specific content of digital literacy to help readers better understand it. In addition, the purpose, motivation and innovation of this paper should be further emphasized.

2. In Section 3, please explain the model and functional equations in detail so that the reader can better understand the research methods.

3. For functional equation (1), please explain the meaning of "" in the equation.

4. Improve the clarity of the bar chart in Figure 1.

5. Literature should be updated to add the latest research results of famous journals in the past three years.

Reviewer #4: The identification strategy needs to be revised to disentangle the effect of the internet on education and access to measure its effects on wages. The paper study an interesting topic but needs a better empirical strategy.

Author's Response:

Thank you very much for your valuable comments on my article. In response to your comments, I have made the following changes, and I would be grateful for your criticisms and corrections.

This paper uses data on Internet use preferences multiplied with Educational attainment and Frequency of study in leisure time to derive the interaction term Improve Digital literacy (IDL) for all three. Through ZINB regression and CMP regression, the results show that enhancing the education of digital literacy and frequency of self-study related to Internet technology has a significant contributing effect on enhancing IDL (Table 8).

In addition, the influence of education level on the labour income effect of individual Internet use preference was clarified by building a mediation effect model.

---

## [Decision Letter · Decision Letter 1]

26 Mar 2024

PONE-D-23-30708R1The Impact of Internet Usage Preferences on Labor Income : Evidence from ChinaPLOS ONE

Dear Dr. Zhang,

Thank you for submitting your manuscript to PLOS ONE. After careful consideration, we feel that it has merit but does not fully meet PLOS ONE’s publication criteria as it currently stands. Therefore, we invite you to submit a revised version of the manuscript that addresses the points raised during the review process.

We look forward to receiving your revised manuscript.

Kind regards,

Yasuko Kawahata

Academic Editor

PLOS ONE

Reviewers' comments:

Reviewer's Responses to Questions

**Comments to the Author**

1. If the authors have adequately addressed your comments raised in a previous round of review and you feel that this manuscript is now acceptable for publication, you may indicate that here to bypass the “Comments to the Author” section, enter your conflict of interest statement in the “Confidential to Editor” section, and submit your "Accept" recommendation.

Reviewer #3: All comments have been addressed

Reviewer #4: (No Response)

2. Is the manuscript technically sound, and do the data support the conclusions?

Reviewer #3: Partly

Reviewer #4: Partly

3. Has the statistical analysis been performed appropriately and rigorously? 

Reviewer #3: Yes

Reviewer #4: No

4. Have the authors made all data underlying the findings in their manuscript fully available?

Reviewer #3: Yes

Reviewer #4: No

5. Is the manuscript presented in an intelligible fashion and written in standard English?

Reviewer #3: Yes

Reviewer #4: Yes

6. Review Comments to the Author

**Reviewer #3:** The research in this paper is very interesting, but it needs more theoretical support and empirical analysis. Such as,

1. For Section 3.3. ① It is suggested to add the description of the solution method of formula (2). The LS method is mainly used in this paper. Is there any other method? Hope to see innovative ideas in the solution method. (2) It is suggested that the convergence conditions of the algorithm and how to deal with different dimensional parameters should be appropriately supplemented; ③ beta in formula (2) should have a subscript.

2. For Part 4, "Empirical Analysis". ① Increase data source analysis and explanation; ② It is recommended to introduce the content and purpose of experimental verification first; (3) Redraw Fig1, which expresses relatively limited information; ④ Add graphical comparison to some tables, as shown in Table 5, which can be displayed with a three-dimensional bar chart to make it more intuitive.

**Reviewer #4:** The paper is still in the preliminary stage. I suggest doing some extra empirical analysis:

1. Improve the descriptive analysis of the variables used; correlations among the independent variables would help to understand the descriptive analysis before the estimation.

3. Clarify the internet usage within the Chinese regions, is it the same across the whole country? Is internet usage similar? Or is it better to collapse similar types to increase the observations in the estimation? IUF, OSN, OSP, ORP, OE, AOI, and OB. It only mentions which has a more considerable use but does not mention the dispersion.

4. Improve the instrumental variable analysis. There is only one instrument, and its validity is not tested. It is necessary to argue why this is the best instrument. The independent variables are likely correlated to the instrument “provincial internet penetration rate,” which is related to income and probably to internet usage. More serious is the fact that the correlations with the unobservable variables related to income are not considered in the estimation, and the instrument does not account for these effects.

5. The authors must show the instrumental variable test to validate their instrument and propose other instruments.

7. PLOS authors have the option to publish the peer review history of their article (what does this mean?). If published, this will include your full peer review and any attached files.

Reviewer #3: No

Reviewer #4: No

---

## [Author Response · Author response to Decision Letter 1]

10 May 2024

Author's Response to Reviewer 3#

Dear Reviewer 3#，Hello! I have made the following improvements to the article in response to your proposed changes:

(1) A description of the solution to Eq. (2) and the formula derivation process have been added in Section 3.3. The bayes:zinb model was also used for parameterization and comparative analysis.

(2) A covariate, PIUW, which is prone to zero inflation, was added to the model in order to enhance the convergence of the regression analysis.

(3) Subscripts were added to beta in equation (2).

(4) Added the analysis and interpretation of CGSS2017 data sources and their types in Section 3.1, and added information on the range of values and units of the variables in the descriptive statistics table of the variables in Section 3.2.

(5) The purpose and content of the empirical analysis were added in section 4

(6) Redrawn Figure 1 based on the percentage of data from INC.

(7) Graphs were added to sections 4.3 and 4.4 to facilitate comparisons by the reader.

Thank you again for your valuable comments! These comments have refreshed the article!

Author's Response to Reviewer 4#

Dear Reviewer 3#，Hello! I have made the following improvements to the article in response to your proposed changes:

(1) Added analysis and explanation of CGSS2017 data sources and their types in section 3.1;

(2) Information on the range of values and units of the variables was added to the descriptive statistics table of the variables in section 3.2.

(3) In section 3.3 the variables were analyzed for correlation using Spearman and Pearson methods.

(8) A description of the solution to Equation (2) and the formula derivation process was added in Section 3.3. The bayes:zinb model was also used for parameterization and comparative analysis.

(9) In section 4.5, “regional cable length” is chosen as a new instrumental variable, which passes the correlation and exogeneity tests; regression analyses are carried out with the help of the cmp model and the instrumental variables, and finally the marginal effects of IUF, OSN, OE, AOI and OB on the improvement of individual labor income are explained respectively.

The conclusions of the article are solidified by your valuable comments! Thanks again for your hard work!

---

## [Decision Letter · Decision Letter 2]

29 May 2024

PONE-D-23-30708R2The Impact of Internet Usage Preferences on Labor Income : Evidence from ChinaPLOS ONE

Dear Dr. Zhang,

Thank you for submitting your manuscript to PLOS ONE. After careful consideration, we feel that it has merit but does not fully meet PLOS ONE’s publication criteria as it currently stands. Therefore, we invite you to submit a revised version of the manuscript that addresses the points raised during the review process.

We look forward to receiving your revised manuscript.

Kind regards,

Yasuko Kawahata

Academic Editor

PLOS ONE

Journal Requirements:

Reviewers' comments:

Reviewer's Responses to Questions

**Comments to the Author**

1. If the authors have adequately addressed your comments raised in a previous round of review and you feel that this manuscript is now acceptable for publication, you may indicate that here to bypass the “Comments to the Author” section, enter your conflict of interest statement in the “Confidential to Editor” section, and submit your "Accept" recommendation.

Reviewer #3: All comments have been addressed

Reviewer #4: All comments have been addressed

2. Is the manuscript technically sound, and do the data support the conclusions?

Reviewer #3: Yes

Reviewer #4: Yes

3. Has the statistical analysis been performed appropriately and rigorously? 

Reviewer #3: Yes

Reviewer #4: Yes

4. Have the authors made all data underlying the findings in their manuscript fully available?

Reviewer #3: (No Response)

Reviewer #4: Yes

5. Is the manuscript presented in an intelligible fashion and written in standard English?

Reviewer #3: Yes

Reviewer #4: Yes

6. Review Comments to the Author

Reviewer #3: There are some problems in the format of this article.

(1) Formula layout, such as formula (3), (5), (8).

(2) Empty pages.

(3) Add chart results and reason analysis.

Reviewer #4: The paper looks better presented. I suggest that the authors must define the abreviations in every table, so they are self-contained.

7. PLOS authors have the option to publish the peer review history of their article (what does this mean?). If published, this will include your full peer review and any attached files.

Reviewer #3: No

Reviewer #4: No

---

## [Author Response · Author response to Decision Letter 2]

6 Jun 2024

Author's Response to Reviewer 3#

Dear Reviewer 3#，Hello! I have made the following improvements to the article in response to your proposed changes:

(1) Checked all formulas and modified the layout of formulas (3) (5) (8).

(2) Adjusted page formatting of Tables 2 and removed blank pages.

(3) Added explanation of Figure 2 in 4.3.

(4) Modified the style of Figures 3 and 4 in 4.4 and added their explanations.

(5) Checked all references and replaced items 21, 29 and 59 of them.

Thank you again for your efforts!

Author's Response to Reviewer 4#

Dear Reviewer 4#，Hello! I have made the following improvements to the article in response to your proposed changes:

(1)Each acronym is defined in Tables 2 in 3.4.

(2)Each acronym is defined in Tables 3, 4 and 5 in 4.2.

(3)Each acronym is defined in Table 6 in 4.3.

(4)Each acronym is defined in Table 7, and Table 8 in 4.4.

(5)Each acronym is defined in Table 9, 10, and Table 11 in 4.5.

(6)Each acronym is defined in Table 12 in 4.6.

(7)Each acronym is defined in Table 13 in 4.7.

(8)Checked all the references and replaced items 21, 29, 59 of them.

 Thank you again for your efforts!

---

## [Decision Letter · Decision Letter 3]

23 Jun 2024

PONE-D-23-30708R3The Impact of Internet Usage Preferences on Labor Income : Evidence from ChinaPLOS ONE

Dear Dr. Zhang,

Thank you for submitting your manuscript to PLOS ONE. After careful consideration, we feel that it has merit but does not fully meet PLOS ONE’s publication criteria as it currently stands. Therefore, we invite you to submit a revised version of the manuscript that addresses the points raised during the review process.

We look forward to receiving your revised manuscript.

Kind regards,

Yasuko Kawahata

Academic Editor

PLOS ONE

Journal Requirements:

Reviewers' comments:

Reviewer's Responses to Questions

**Comments to the Author**

1. If the authors have adequately addressed your comments raised in a previous round of review and you feel that this manuscript is now acceptable for publication, you may indicate that here to bypass the “Comments to the Author” section, enter your conflict of interest statement in the “Confidential to Editor” section, and submit your "Accept" recommendation.

Reviewer #3: All comments have been addressed

Reviewer #4: All comments have been addressed

2. Is the manuscript technically sound, and do the data support the conclusions?

Reviewer #3: Yes

Reviewer #4: Yes

3. Has the statistical analysis been performed appropriately and rigorously? 

Reviewer #3: Yes

Reviewer #4: Yes

4. Have the authors made all data underlying the findings in their manuscript fully available?

Reviewer #3: Yes

Reviewer #4: Yes

5. Is the manuscript presented in an intelligible fashion and written in standard English?

Reviewer #3: Yes

Reviewer #4: Yes

6. Review Comments to the Author

Reviewer #3: The format of the paper needs further improvement，such as

1.Formulas (3) and (5) are not typeset correctly.

2.Formula (5) is followed by an extra empty page.

3.The three-line table format is not correct. The first and third lines are thicker than the second line.

4.The labels for the horizontal and vertical axes of the graph should be complete, please check and complete them if necessary.

5.Figure 1 is not beautiful, especially the thickness of the horizontal and vertical coordinates is not consistent.

Reviewer #4: The authors addressed the previous comments. The paper is solid. The authors might consider showing the percentage of use of OSN, OSP, ORP, OE, AOI, and OB instead of presenting the mean in Table 1 because they are not continuous variables but ordinal variables.

7. PLOS authors have the option to publish the peer review history of their article (what does this mean?). If published, this will include your full peer review and any attached files.

Reviewer #3: No

Reviewer #4: No

---

## [Author Response · Author response to Decision Letter 3]

5 Jul 2024

Author's Response to Reviewer 3#

Dear Reviewer 3#，Hello! I have made the following improvements to the article in response to your proposed changes:

1. The layout of formulas (3) and (5) has been re-edited.

2. There is no blank page after the adjustment of formula (5).

3. The first and third lines of the three-line table have been bolded.

4. The labels of the horizontal and vertical axes of the charts have been completed.

5. The graph has been redrawn to solve the problem of inconsistent thickness of horizontal and vertical coordinates.

Thank you again for your efforts!

Author's Response to Reviewer 4#

Dear Reviewer 4#，Hello! I have made the following improvements to the article in response to your proposed changes:

Figure 1 was plotted and explanatory text added to the article to explain the status (in per cent) of survey respondents' Internet usage preferences.

 Thank you again for your efforts!

---

## [Decision Letter · Decision Letter 4]

19 Jul 2024

The Impact of Internet Usage Preferences on Labor Income : Evidence from China

PONE-D-23-30708R4

Dear Dr. Xiaoxia Zhang,

We’re pleased to inform you that your manuscript has been judged scientifically suitable for publication and will be formally accepted for publication once it meets all outstanding technical requirements.

Kind regards,

Yasuko Kawahata

Academic Editor

PLOS ONE

Additional Editor Comments (optional):

Reviewers' comments:

Reviewer's Responses to Questions

**Comments to the Author**

1. If the authors have adequately addressed your comments raised in a previous round of review and you feel that this manuscript is now acceptable for publication, you may indicate that here to bypass the “Comments to the Author” section, enter your conflict of interest statement in the “Confidential to Editor” section, and submit your "Accept" recommendation.

Reviewer #3: All comments have been addressed

Reviewer #5: All comments have been addressed

Reviewer #6: All comments have been addressed

2. Is the manuscript technically sound, and do the data support the conclusions?

Reviewer #3: Yes

Reviewer #5: Yes

Reviewer #6: Yes

3. Has the statistical analysis been performed appropriately and rigorously? 

Reviewer #3: Yes

Reviewer #5: Yes

Reviewer #6: Yes

4. Have the authors made all data underlying the findings in their manuscript fully available?

Reviewer #3: Yes

Reviewer #5: Yes

Reviewer #6: Yes

5. Is the manuscript presented in an intelligible fashion and written in standard English?

Reviewer #3: Yes

Reviewer #5: No

Reviewer #6: Yes

6. Review Comments to the Author

Reviewer #3: I think the paper is acceptable. It is recommended to carefully check the typography of all formulas and make them as consistent as possible.

Reviewer #5: The paper "Impact of Internet Usage Preferences on Labor Income: Evidence from China" aims to empirically analyze micro-level survey data to reveal the impact of individual differences in internet usage preferences on their labor income.

The abstract contains unnecessary information, such as mentioning that relevant data from the authoritative Chinese General Social Survey (CGSS2017) were selected. This can be cut down to make it easier for the reader.

Why were three models compared?

Reviewer #6: I have carefully reviewed this manuscript and below is my decision.

The paper is well-suited for publication in the Plos One. All comments are well-improved the quality of paper. Good luck!

7. PLOS authors have the option to publish the peer review history of their article (what does this mean?). If published, this will include your full peer review and any attached files.

Reviewer #3: No

Reviewer #5: No

Reviewer #6: No

---

## [Editor Report · Acceptance letter]

26 Jul 2024

PONE-D-23-30708R4 

PLOS ONE

Dear Dr. Zhang, 

I'm pleased to inform you that your manuscript has been deemed suitable for publication in PLOS ONE. Congratulations! Your manuscript is now being handed over to our production team.

Kind regards, 

on behalf of

Dr. Yasuko Kawahata 

Academic Editor

PLOS ONE